# Living Fatherhood in Adults Addicted to Substances: A Qualitative Study of Fathers in Psycho-Rehabilitative Drug Addiction Treatment for Heroin and Cocaine

**DOI:** 10.3390/ijerph17031051

**Published:** 2020-02-07

**Authors:** Pasquale Caponnetto, Chiara Triscari, Marilena Maglia

**Affiliations:** 1Departement of Education, University of Catania, 2 Ofelia, 95124 Catania, Italy; chiara.triscari996@gmail.com (C.T.); marilena.maglia@gmail.com (M.M.); 2Center of Excellence for the Acceleration of HArm Reduction (COEHAR), Dipartimento Di Medicina Clinica E Sperimentale, University of Catania, 95131 Catania, Italy

**Keywords:** addiction, psychotherapy, qualitative study, substance abuse, father

## Abstract

The research aims to investigate the emotional experiences of the individuals who use drugs in terms of their parenting role as father, within a rehabilitative context. The study aims to analyze how dependence affects the exercise of the parental role, specifically paternity, with the aim of offering an overview of the father-son relationship while considering the possible limitations that characterize those who are forced to live parenting in an atypical way. It is necessary to help individuals who use drugs to be able to help them in life with the goal of establish a better parenting awareness and a good relationship with their father and their children. The research work made use of qualitative tools, specifically semi-structured interview, which was administered to a sample of 18 fathers that were treated in a rehabilitation clinic for individuals who use drugs. The used semi-structured interview made it possible to analyze the perception of participants about their paternity, the quality of the relationship with their father and their children, and the influence that the narcotic substance has generated in the relationship with their father and with their children.

## 1. Introduction

The father’s role in the care and development of his children is very important, in fact during infancy, fathers have been shown to be competent and capable of positive interactions with young infants to have similar psychological experiences as mothers [1]. Studies showed that the father is more likely to be the infant’s play partner than their mother, and father’s play tends to be more stimulating, vigorous, and arousing for the infant [2,3].

Paternal involvement in the early childhood years is associated with positive child developmental and psychological outcomes over time, while, during adolescence, several recent national longitudinal studies have shown that father involvement is associated with a decrease in the likelihood of adolescent risk behaviors [4,5] and predicts less adolescent depressive symptoms for both genders [6].

A study showed that fathers who use drugs might play a more limited role in their children’s lives and provide less support than non-substance-consuming fathers [7].

Several studies showed that fathers who use drugs are more likely to be aggressive with parents [8], lower sensitivity and empathy [9], poor problem solving skills and higher rates of negativity during interactions [10], and more problematic and less adequately supervised their children [11].

Being relevant to these results, mood disorders seem to mediate the association between drugs use and hostile-aggressive parenting and paternal sensitivity [12,13], which suggests that the assessment and treatment of mood disorders may be important for improving parenting for men who consume drugs.

There is also evidence of overlapping drugs use, intimate partner violence (IPV), and mistreatment [14,15,16] with children in people who are affected by substance use disorders. In addition, there is some emerging evidence that, in the context of drug use, fathers who report IPV also report more hostile-aggressive parents [17]. It is not drug use, but rather the propensity for hostility and the use of violence that is associated with hostile-aggressive and abusive parenting behavior [18]. Several studies have shown that fathers with substance use problems report lower paternal satisfaction, increased feelings of guilt for their parenting, and concern regarding the type of role model that they have been for their children [19,20].

A survey of parents in outpatient treatment for drugs use reported that they had concerns regarding their children, and the majority of participants said that they would benefit from a class and they would like paternity or parenting substance treatment issues [21]. A study of fathers in residential substance use disorder treatment showed that 95 percent of participants always thought about their children, 77 percent declared that they would be interested in paternity and co-parenting sessions as part of their residential treatment [22]. These studies suggested these men are open in interventions that would benefit their relationship with their children.

Wanting to give a definition of the paternity of the individuals who use drugs, it is possible talk about unacknowledged paternity and deserted paternity [23]. Misrecognition derives from the negative perception, by the subject and from the family environment, that this in its vulnerable condition might be capable of positively fulfilling the paternal function, while desertion derives from the guilt and inadequacy of the toxic subject towards his own person, his life, and his choices, which leads him to live the paternity intermittently [24]. All of this could lead the people who use drugs, in reference to the possibility of being a father, who could abort his function, choosing to give up any responsibility that derives from estranging himself from the relationship with his children and repressing within himself the suffering that follows, or even be excluded from his function by the wife, which, by amplifying the negative elements, would arouse a sense of conflict in the child towards the father. The “resigning” father, as defined by Lucarini, due to lack of guilt or laziness, is unable to leave the child with his own sign, or that of the wound resulting from the loss [25].

Children without a father had lower capacity for defining the sexual role, greater hostility, lower academic success, and greater relational difficulties [26]. The father plays a decisive role, with the child since the first contact with paternal masculinity this process makes a first differentiation of the sexes, important for the mental representation that will be used as an orientation model regarding the expected social behaviors construction [24]. The other elements cited can be traced back to the failure to meet the moral norm, being embodied by the father during child development, which educates, corrects, and obliges the child to abandon the original condition of omnipotence. This process allows for the child to put order in the primitive internal chaos and open up to the relationship, without anxiety. It is also possible to state that the effects of paternal deprivation are not exclusively directed to children, but they involve the whole family system [27], since the members are often called upon to become substitute models to fill the missing position left by him. 

The following qualitative analysis that was carried out on a group of participants treated at a therapeutic community that is a participative, group-based approach for long-term drug addiction rehabilitation with the aim of detecting how the physical or emotional paternal absence was widely present or not in the life of those who use substances.

It has been decided to carry out research with the double objective of exploring the phenomenon of paternity in a condition of drug addiction rehabilitation through the eyes of those who live this experience every day, and to explore if and how much it affects the parent-child relationship.

## 2. Material and Methods

### 2.1. Research Questions

The research questions of this qualitative study will explore the following areas:the relationship of the participants with their father and with educational models; and,fundamental values attributed to the father figure;the effects of substance abuse in the father-child relationship.

### 2.2. Participants

This qualitative research was conducted with a sample consisting of 18 males, in psycho-rehabilitative drug addiction (heroin and cocaine) and residing in a therapeutic community. All of the participants completed a written consent form for inclusion before they participated in the study. The study was conducted in accordance with the Declaration of Helsinki, and with the GDPR requirements. The study was conducted in agreement with the ethical norms set by the Italian National Psychological Association.

### 2.3. Metodology

In this study, a qualitative approach was used that allows us to explore the point of view of the father who lives the relationship with his child within a therapeutic context through the personal description of their experiences and their perspectives [28].

Qualitative research is a method of enquiry, which is based on the report of events gained through observation or interaction with participants, to gain an in-depth understanding of human behavior. It draws from the situation in which events occur and attempts to describe occurrences, as a means of determining the process in which events are embedded and the perspectives of those participating in the events, while using induction to derive possible explanations based on observed phenomena’ [29].

Following what is expected from this trend methodology, a semi-structured interview was built, consisting of 18 questions. The questions allowed for gathering information about:general characteristics of the participants;the type of family in which they recognize themselves;their perception of paternity;the relationship with one’s father;the absence of the father figure in their lives and during the period of dependence;the perception of self as a father;the trust placed in the educational role of the partner;the fundamental characteristics in terms of behavior, attitude, and personality, which should absolutely have the father figure; and,the character and personological aspects that a father should not have.whether and how addiction to substances has affected the parent-child relationship.

### 2.4. Eligibility Criteria

Participants were eligible for the study if they;

(1) Had a diagnosis of substance abuse disorders according to DSM-V criteria.

(2) Father in psycho-rehabilitative treatment and residing in a therapeutic community.

(3) No history of major depression or other psychiatric conditions.

(4) Certification of substances use disorders (heroin and cocaine) for at least six months.

### 2.5. Sampling and Recruitment Strategy

In this study, two non-probability sampling techniques were used, namely purposive sampling and quota sampling, which complemented each other. Purposeful sampling is widely used in qualitative research [30]. In this type of sampling, participants are selected or sought after according to pre-selected criteria based on the research question (i.e., the eligibility criteria). Quota sampling is a sampling technique, whereby a participant quota is pre-set prior to sampling [30] (a number of 18 fathers). The sample size was predetermined: 18 fathers with were recruited. According to Creswell [31], five to 25 participants are required for qualitative studies.

### 2.6. Setting

The research was conducted at the therapeutic communities. During the course of the study, it was possible to administer the interview to participants in rehabilitation treatment in two communities on two separate days supported by the presence of internal staff.

### 2.7. Data Collection and Processing

Individual one-to-one semi-structured interviews were used to collect data, conducted by two trained clinical psychologists. Interviews took place during working hours at the Rehab Center in a private room, where no one else was present; Participants were given the option to have their recovery worker with them; all of the interviews were recorded. For each interview, a space called the “opening phase” was dedicated, which involved researchers that aimed to help the participant feel comfortable using arguments about everyday life. The duration of the interviews ranged from 30 min. to one hour and the average length of interviews was approximately 45 min. (some participants became distressed or were easily fatigued, which influenced the length of the interview). All of the verbatim interviews were transcribed within a week of each interview. The interview question has been piloted. The involved researchers took an average of two to four hours to transcribe each interview script. Many of the participants had chosen to use the Sicilian or Italian and Sicilian language during the interviews, which made the transcription and translation complex at times, mainly because it was important to keep the information that was presented by participants.

### 2.8. Data Management

The data were organized in a systematic way while conducting the work in the field. Recording field notes every day, transcribing interviews, and attaching memos accordingly. NVivo 11, a Computer Aided Qualitative Data Analysis Software (CAQDAS) program, was used to manage the data. The use of the NVivo software (QSR International, Melbourne, Australia,) facilitated the process of organizing, re-arranging, and managing the considerable amount of data. The interview transcripts were formatted in Microsoft Word to facilitate importing the transcripts into N-Vivo. When importing the transcript into NVivo, this resulted in the questions being displayed in the content panel in the N-Vivo explorer. Hence, when selecting a question, it was possible to jump to this section in the interview transcript. After coding the interviews in NVivo, all of the passages assigned to a specific code were viewed on screen and printed. Each participant’s biographical data was accessed and these were kept under a knot classification file. 

### 2.9. Analysis

In this study, a thematic analysis involving multiple angles and interactions of participants’ opinions was used. Braun & Clarke [32] describe thematic analysis as “a method for identifying, analyzing, and reporting patterns (themes) within data” [32]. It is also described as a flexible and useful research tool that provides a rich and detailed, yet complex, account of the data [32]. Thematic analysis involves the search for and identification of common threads that extend across an entire interview or set of interviews [33] and a ‘theme’ is the main product of data analysis [34]. Following the six-phased guide that was suggested by Braun & Clarcke [32], the first step in thematic analysis involves becoming closely familiar with the data by reading and re-reading the interview transcripts. The transcriptions were studied one at a time by involving the text through highlighting and memorization the different sections, and then generating 1–2 pages summaries on each interviews. Following the close reading phase, independent notes were taken by the second psychology and both researchers developed a coding system that included deductive codes derived from the thematic guide and inductive codes that emerged from the interview data. Using NVivo software, sections of text were encoded in the transcript that aligned with the code descriptions and the surrounding context. In this process, the researchers used codes as filters for large amounts of text data and organized sections of text around a common description. The codes were rearranged into themes once the codes were applied to all transcripts. In thematic analysis, a theme is defined as ‘a pattern found in the information that at a minimum describes and organizes the possible observations and at a maximum interprets aspects of the phenomenon’ [35]. Three broad themes were identified and related to the original research questions: (1) the relationship of the participants with their father and with educational models; (2) fundamental values that are attributed to the father figure; and, (3) the effects of substance abuse in the father-child relationship. The final phase can be described as the process of synthesis [32]. Once each theme was clearly defined and described and the theme was illustrated with reference to transcriptions while using excerpts of verbatim quotations of the study participants to capture the essence of the theme.

## 3. Results

This part starts with a description of the participant’s characteristics, and then goes on to give a tabulated overview of the themes followed by a more detailed synthesis and discussion.

### 3.1. Participants’Characteristics

Eighteen interviews were conducted with 18 males (Table 1). This qualitative research was conducted with a sample of participants in psycho-rehabilitative treatment for substance abuse disorders (heroin and cocaine) residing in a therapeutic community. The participants were all of Italian nationality, aged between 21 and 56 (median age 38.5) and with a degree that varies between middle school and high school diploma, and only one subject has completed the fifth grade. Regarding the marital status, two declared to be divorced, two separated, seven identified themselves as single inside the family nuclear system, two as single inside an extended family, and five as single inside a the de facto family. 

### 3.2. Themes

Participants’ views were synthesized into three themes and related subthemes (Table 2).

#### 3.2.1. Theme 1: Relationship with the Father

From a careful reading of the answers given, it emerged that the participants developed a greater attachment to the maternal figure, possibly as a consequence of the peripheral position taken by the father during their growth. This hypothesis is supported by the answers given to the question ‘what kind of father was yours?’ to which majority of the participants replied that they had a father who was not present in their life or present only as an economic support figure. Nevertheless, it was found that the participants maintained a positive view of fatherhood and a strong motivational drive to improve self-improvement as a father, despite the fragility of self-perception as a father.


*“My father was absent and disinterested, or at least unable to interact positively” (Antonio., 37 years old)*


##### Subtheme 1: Paternal Family Script

The first sub thematic refers to family scripts, that is, to all of those mental representations of an attachment behavior that structures an individual’s personality and influences the way they act over time. Although many subjects had previously stated that they did not see their father as a role model for the education of themselves, they later replied that they noticed similarities between them as fathers and the father figure with whom they related during childhood. This factor leads us to support the importance of having stable educational figures of reference, as these, although implicitly, will affect the parenting style adopted and reflect the essence of the family of origin. Therefore, there seems to be a kind of indirect identification that affects their lives, since the pathological relationship with the substance in their daily life, in turn, produces that much rejected physical and emotional detachment. The understanding of these dynamics leads me to hypothesize the existence of what Andolfi [36], called a trigenerational bond, according to which the ability to be in the world does not end in communication transitions and affective flows, but involves revealing the existence of an individual substructure that binds members of the family system [37].


*“Yes, in some aspects I find similarities with my father and my way of being a father” (Alfio., 48 years old)*


##### Subtheme 2: Paternal Deficiency

Most of the participants said that they felt their father were distant, and that they did not consider him an educational model of reference because he was “too strict”, “little present”, “because he was not seen me as often drunk”, or “because he did not know how to behave”. These answers raise possible hypothesis for us that the absence of a balanced paternal reference model, inclusive and the overwhelming emptiness left by its lack, could be determining factors of substance intake, sought with the purpose of changing that condition of inner imbalance fed over time.


*“I have always looked for my father, but he has always been too absent for different work commitments, for too much age difference and for the presence of a double family” (Carmelo., 39 years)*


#### 3.2.2. Theme 2: Values Associated with the Father

In the definition of these criteria, the researcher, precisely to allow for the participants of the research to find and express the adjectives that, in their opinion, most qualify this figure, inserted no guidance adjective. When asked “what characteristics of behavior, attitude, and personality are absolutely fundamental, in your opinion, in a father figure” the most widely used adjectives were: respect, presence, understanding, responsibility, openness to dialogue, support, authority, patience, education, affectivity, and regulation. On the other hand, exploring “the aspects that a father should not have” was found to have a strong incidence of adjectives such as: absolute authority, severity, rigidity, violence, indifference, affectivity, and aggression towards his wife and children. From this analysis emerges the tendency to describe and search for a father figure in line with the disappointed expectations, that is, a figure that is capable of sustaining with his own presence, love, and capacity to welcome the family structure, and the relationships that are part of it.


*“A father must have patience, wait for us to be ready, to be able to support for good and evil, to have a constructive way of making things understood, he must not raise his hands, he must not transmit anxiety and think for himself and his children” (Gaetano., 43 years)*


##### Subtheme 1: Change in the Father Role

From the previous analysis focused on the perception of the father figure, a sub thematic emerged in the sharing of the idea of a transformation of the paternal parenting role over time. This transformation is perceived by the participants in a positive way, since, when compared to the past, the father is thought to be a more collaborative figure in the division of parenting tasks, despite the persistent difficulties in relation to the figure and more present in the lives of children. These data, although not directly linked to the core of the research, have contributed to a broader view of the paternal idea widespread today.


*“Today the father is more present and less regarded as a figure of economic support” (Andrea., 50 years)*


##### Subtheme 2: Presence, Understanding and Authority

After outlining through interviews the perception of general and subjective paternity for the participants, through a more in-depth analysis exploring the value elements attributed to the father figure, the interviews, in line with the previous sub thematic, revealed a broad agreement among the participants regarding the paternal presence, understanding, and authority. Therefore, the participants highlighted the importance of a collaborative and supportive parenting figure, a detente of limits, but also a container of emotions. The father is therefore perceived as an essential figure that is capable of psychologically and morally forming the children.


*“The father must be strict at the right point, help for the proper education of his children but the most important thing is that he must also be a friend for his child, he must not be violent or absent, must be respected for the good he gives and for his presence” (Davide., 30 years)*


#### 3.2.3. Theme 3: Addiction and Paternity

The last theme emerged by asking fathers in a rehabilitative context if and how the substance affected the relationship with their child. The answers analyses showed that substances act as a kind of “separated” in the father-son relationship, since, as a father has said, *“it’s moving away and destroying, with a word it destroys relationships”*. The majority of substances-addicted fathers spoke of their fatherhood in terms of distance *“I have practically seen my children grow up in my absence”*, *“My daughter, given my addiction, was afraid to be alone with me because of my manias of persecution and still does not trust in me”*, *”Substances addiction as well as ruining my person has ruined the growth of my children”.* The inability of the substances-addicted parent to fail to embody the present, supportive, and responsible figure of their imagination as outlined in the previous paragraph, translates into terms of shame. Therefore, this condition urges them, as far as is possible, not to reveal their problem to their children. In conclusion, addiction negatively affects the relationship with children, undermining the relational field. These fathers said they felt more *“overwhelmed”* when compared to other fathers, *“sometimes indifferent”, “father”,* and *“absent”.*


*“My addiction affected my fatherhood. My addiction affected my relationship with my daughter because I miss her, she cries when she comes to see me and then she has to leave me. I love her so much. (Angelo, 43 years)*


##### Subtheme 1: Addiction and Family Relationships

The research has allowed us to analyze how individuals experience fatherhood in an addiction situation, but it has provided us with a broader picture, allowing for us to also explore the quality of relationships held with other family members. Interviews have shown that the sense of estrangement, emotional fluctuations, and the extreme concentration of the dependent individual who use drugs on himself induce him to live relationships with indifference and superficiality. It is possible affirm that addiction acts as a destructive factor within the family environment, negatively affecting all members.


*“Dependence has alienated me, it is destructive, with a word it destroys relationships” (G., 43 years)*


##### Subtheme 2: The Children’s Experiences

The last subtheme is related to the perception of paternal dependence from the point of view of children. Participants, in addition to defining what aspects of the father-child relationship compromised by the addiction problems, tried to describe the psychological condition of their children. These parents highlighted the state of non-acceptance of the paternal condition by their older children and the futile attempt to deny their state to their youngest children to preserve them from suffering. Despite this attempt, these parents have stated that they have difficulty in serving as a caring father and in nurturing their relationship with them, as they are often wary of them, distant, and forced to confront realities suitable for them.

## 4. Discussion

This research has set out to explore dependent fatherhood in rehabilitative treatment for substance addiction experience, their status as fathers, and how their addiction has an influence on the parent-child relationship. A similar qualitative study explored the experiences of father in relation with their children and concluded that is important to promote the quality of treatments that are delivered to individuals who use drugs by using family therapy and psychotherapy, because this reduces their problems and the ones they have with their children [38].

One of the issues that emerged highlighted was that of parenting, in which the study participants said they felt insecure and disoriented, as their addiction limits the exercise of parenting function, preventing them from being physically and psychologically present during the growth of their children. This leads to some questions regarding fatherhood, atypical forms of parenting (including that of the individuals who use drugs), and the consequences that they may have within the family environment. The initial hypothesis looked at how substance addiction had an emotional and psychological impact on the exercise of parental function, making the father-child relationship more fragile.

A study that was conducted by Collins and collaborator [39] explored the parental role of individuals who use drugs and found that 51% of their study sample was considered to be highly involved with their children and showed lower levels of addiction severity compared with less involved fathers.

An initial analysis was carried out in relation to the scientific landscape of drug addiction, paternity, and dependent parenting. Within the landscape of psychological studies, the theme of fatherhood was taken into account little, since, in the collective imagination, the figure of the father was considered to be distant from the emotional and communicative contact with his children. An in-depth analysis of the role of parenting has allowed exploring the father figure from the point of view of the children, showing that “children demand the presence of the father” [25] and it is fundamental to a proper psycho-affective development. This research has highlighted the importance of caring for the family in the care path, since “every single part only makes sense in relation to the others” [40].

These individual who use drugs can be encouraged to share the negative impact of their addiction on their family system and on themselves support them to share their emotions that are linked to their family experiences is important, as it helps them to reframe their self. In such contexts, it is necessary to focus attention not only on the individual person that is affected by pathological dependence but also on the family context of the same current is of origin. Observing these contexts allows for us to know if and what dynamics can exist within the family and what consequences it has had on the individual person. Therefore, in these cases, it is useful to establish family gatherings that can help not only the individual who uses drugs, but also the family members in the management of the crisis and in re-educational interventions [41].

The impact of the absence of a stable paternal reference in the life of the subjects is not insignificant, as, in line with Bowen’s trigenerational theory [27], should induce operating on the whole family, in order to prevent children from being able to reliving the suffering experienced by his father. In this study limits were found, including the small number of participants in the research, while considering that not everyone in the therapeutic community was fathers, the impossibility of interviewing the family members of the participants and finding a sample of non-dependent fathers with age, social context, and level of education similar to that of participants with whom to compare data, to understand if there are variations in the perception of fatherhood, and whether paternal absence has an impairment in the parental role even in those considered “individual who doesn’t use drugs”. However, these limitations could be a starting point for further research.

## 5. Conclusions

In conclusion, it is possible to say that the paternal role is fundamental in the growth of children, who live a condition of parenthood hovering between the desire to enter into a healthy relationship with their children and the impossibility of living the quotidianity with the same. This ambiguity leads, especially in cases of long rehabilitation of fathers in residential contexts, to establishing relational dynamics with their children that are characterized by psychosocial vulnerability and inadequate or distorted family communication.

Therefore, in these contexts, it is considered to be necessary to increase the psycho-rehabilitative network by providing a more predominant clinical support that is useful to all fathers who undertake a pathway of treatment for detoxification from substances.

## Figures and Tables

**Table 1 ijerph-17-01051-t001:** Characteristics of participants.

Characteristics of Participants	Total (n = 18)
Gender male: n (%)	18 (100%)
Age: mean years (sd)	42.3 (9.1)
Age onset of substance abuse disorders: mean years (sd)	18.1 (4.8)

**Table 2 ijerph-17-01051-t002:** Emerged Themes and Sub Themes.

Theme	SubTheme
Relationship with thefather	Paternal family script Lack of paternal presence
Values associated with the father	Change in the father role Presence, understanding and authority
Addiction and paternity	Addiction and family relationships The children’s experiences

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
