# Peer review of "Living Fatherhood in Adults Addicted to Substances: A Qualitative Study of Fathers in Psycho-Rehabilitative Drug Addiction Treatment for Heroin and Cocaine"

_ijerph, 2020, doi:10.3390/ijerph17031051_

Round 1

Reviewer 1 Report

The manuscript requires massive review of grammar and style. The manuscript should be written in the passive form i.e. try to avoid using “I” did this. The manuscript contains numerous typos and words/sentences without spaces. Results and discussions need to be better presented. Discussion is superficial and requires support from the literature.

Currently, the term “drug addict” is now deemed a term that increases stigma and a barrier to treatment. So we are encouraged to choose alternative terms such as “individuals who use drugs”.

Abstract: lines 14-17 – sentence too long and difficult to understand. Please review grammar.

Line 57: “the” is repeated twice

Line 75 and 78, 79, 98, 102, 134, 149, 182, 196, 301, Table 1 and 2: please add spaces between words and sentences

Line 91: typo

What are the therapeutic communities?

What’s a node classification file?

Lines 97-98: please add a reference

Methodology: please provide information on client consent provision, how were GDPR requirements met? Ethics protocol ID number

Please provide more information on the interviews: I understand that the interviews were carried out with individual patients (one-to-one interviews). Were clients given the option to have their recovery worker with them, for example?

How did you validate your interview questions? Have you piloted your interview questions?

Line 137: please remove comma after memoing

Line 141 and 158: informal language

Line 158: starts

Line 164: please add the median age

Line 166: please review grammar

Font size lines 184-185 and 208-210

Typo line 190, 248, 272, 285, 301, 307

Line 261: did you mean to say “she” in that sentence?

Please review references

Author Response

See attached the file.

Thanks for your feedback

Reviewer 2 Report

AUTORES

Conclusions: The objective of the study can be interesting because it tries to contribute knowledge about the paternity related to people with substance use problem in rehabilitation. However, some aspects of this manuscript create important doubts about the interpretation of the results, mainly its theoretical justification and the sample size.

Some specific issues:

Introduction

-           I think too concise. It would be good if the authors delve deeper into the objective of the study and its relevance.

At the end of the introduction, the objectives of the study and the hypotheses from which the authors start are lacking.

Methods

-           The inclusion and exclusion criteria I think may be scarce. Were patients with severe mental disorders included? How long had they been consuming, all kinds of substance? Perhaps with these selection criteria it is not enough to be able to have a homogeneous sample.

Discussion

- This manuscript´s section I think it must be reviewed and completed. There is a lack of an in-depth discussion of the previous literature and the results found in terms of the main objective of the study. In my opinion, the discussion does not make clear what the data mean and their value in the educational field.

Author Response

See attached  file.

Thanks for your feedback

Round 2

Reviewer 1 Report

Thanks for the authors for editing the manuscript and improving the methodology section. The manuscript still require extensive grammatical editing e.g. line 156.

Line 9: please remove "the" before "individuals who use drugs"

Too many full stops on lines 15, 48, 153 etc

Too many spaces or none at all on lines 17, 32, 34, 36, 43, 53, 59, 64, 71, 158, 176 and many more.

Formatting error: section 2.9 in line 156 and section 2.8 in line 167

Reviewer 2 Report

After reading the paper, I consider that the article has improved substantially. The research presented deals with a very interesting topic, such as the need to know more about to help substance abuse people to take up again in their own life to improvement the possible to included again this people on a social life.